# A mathematical model of the impact of insulin secretion dynamics on selective hepatic insulin resistance

Gang Zhao[1], Dagmar Wirth[2,3], Ingo Schmitz [4,5] & Michael Meyer-Hermann[1,6]

Physiological insulin secretion exhibits various temporal patterns, the dysregulation of which is involved in diabetes development. We analyzed the impact of first-phase and pulsatile insulin release on glucose and lipid control with various hepatic insulin signaling networks. The mathematical model suggests that atypical protein kinase C (aPKC) undergoes a bistable switch-on and switch-off, under the control of insulin receptor substrate 2 (IRS2). The activation of IRS1 and IRS2 is temporally separated due to the inhibition of IRS1 by aPKC. The model further shows that the timing of aPKC switch-off is delayed by reduced first-phase insulin and reduced amplitude of insulin pulses. Based on these findings, we propose a sequential model of postprandial hepatic control of glucose and lipid by insulin, according to which delayed aPKC switch-off contributes to selective hepatic insulin resistance, which is a long-standing paradox in the field.

[1] Department of Systems Immunology and Braunschweig Integrated Centre of Systems Biology, Helmholtz Centre for Infection Research, Rebenring 56, 38106 Braunschweig, Germany. [2] Model Systems for Infection and Immunity, Helmholtz Centre for Infection Research, Inhoffenstraße 7, 38124 Braunschweig, Germany. [3] Institute of Experimental Hematology, Hannover Medical School, Carl-Neuberg-Straße 1, 30625 Hannover, Germany. [4] Systems-Oriented Immunology and Inflammation Research Group, Department of Immune Control, Helmholtz Centre for Infection Research, Inhoffenstraße 7, 38124 Braunschweig, Germany. [5] Institute for Molecular and Clinical Immunology, Otto-von-Guericke University, Leipziger Straße 44, 39120 Magdeburg, Germany. [6] Institute of Biochemistry, Biotechnology and Bioinformatics, Technische Universität Braunschweig, Spielmannstraße 7, 38106 Braunschweig, Germany. Correspondence and requests for materials should be addressed to M.M.-H. (email: mmh@theoretical-biology.de)

The liver is a key organ in maintaining whole-body glucose homeostasis. The balance between glucose production, for example during fasting, and glucose storage after feeding is largely controlled by the two counteracting pancreatic hormones insulin and glucagon. The former suppresses hepatic glucose production, while the latter promotes it. Reduced suppression of hepatic glucose production after feeding is a primary defect in impaired glucose tolerance (IGT)[1, 2] and is preceded by abnormal insulin secretion during the course of prediabetes development[1].

Two dynamic features of insulin secretion have been implicated in postprandial glycemic control in prediabetes and type II diabetes (T2D). At first, impaired insulin secretion is characterized by diminished first-phase insulin. After a general mixed meal, plasma insulin concentration increases promptly, peaking at about 30 min[3, 4]. This acute postprandial insulin release, or the first-phase insulin release, suppresses endogenous glucose production in the liver[5–7]. There is evidence that the effects of the first-phase insulin persist even after 2 h: plasma insulin levels 30 min after an oral glucose load are inversely correlated to the plasma glucose concentration at 2 h. Furthermore, first-phase insulin secretion, during both oral glucose tolerance test, and hyperglycemic clamp, showed a negative linear relationship with fasting glucose[8]. In T2D patients, the peaking of the first-phase insulin is weakened and delayed[3, 4]. Rectifying the defect in early phase insulin secretion, for example by exogenous insulin infusion over 30 min at the beginning of a meal while inhibiting endogenous insulin secretion by somatostatin, improved glucose control in T2D patients[9, 10]. In contrast, a constant injection rate of insulin during 3 h did not significantly alter the glycemic control, although the total amount of infused insulin was the same[9, 10]. Similar results were obtained when comparing the effects of subcutaneous insulin and lispro[11], which is a fast-acting insulin analog.

A second impaired feature of insulin secretion in prediabetes and T2D is pulsatile insulin delivery. Portal vein insulin shows an oscillatory behavior with a periodicity of ~5 min[12], which means pre-hepatic first-phase insulin is accomplished by ~6 pulses of insulin. Compared to other insulin target organs, the liver is unique in the sense that it is exposed to insulin pulses of much higher amplitude. In humans and large animals, the amplitude of portal vein insulin pulses is ~100-fold higher than in the systemic circulation[12]. The physiological regulation of insulin secretion, for example by incretins, somatostatin, sulfonylurea and age-related insulin resistance, mainly modulates the amplitude of insulin pulses[13–15]. In T2D, prediabetes, and even glucose tolerant first-degree relatives of T2D patients, the pulsatile delivery of insulin is impaired, in terms of both, the amplitude and the temporal regularity of the pulses[16, 17]. Recent studies with rodents

and canines suggested that the liver is most sensitive to pulsatile insulin, in terms of insulin signaling and its effects in suppressing glucose production[18]. Specifically, pulsatile infusion of exogenous insulin directly into the pre-hepatic vein showed enhanced effects in the activation of key signaling molecules, transcription factors, and ultimately in the suppression of hepatic glucose production, as compared to constant insulin infusion or to a pattern mimicking T2D[18]. Insulin has both direct and indirect effects in suppressing hepatic glucose production[19]. The former is mediated by insulin receptors on the membrane of hepatocytes, and the latter is mediated by effects of insulin in the brain, adipose tissue, muscle, and pancreatic alpha-cells. Given that the systemic levels of insulin, glucagon, and free fatty acids were comparable in the pulsatile, constant and T2D-mimicking infusions, it was presumed that hepatocyte insulin receptor mediated pathways contribute mainly to the observed enhancement in glucose control[18]. Pulsatile insulin infusion into the antecubital vein also showed better glycemia control than constant infusion[20].

Insulin induces auto-phosphorylation of the insulin receptor, which leads to the phosphorylation of tyrosine residues on insulin receptor substrate (IRS) proteins. In the liver, the major IRS proteins are IRS1 and IRS2. Phosphorylated IRS proteins activate multiple signaling pathways, among which Akt and atypical protein kinase C ζ/λ (aPKC) are two key metabolic effectors of insulin[21]. Insulin regulates both carbohydrate and lipid metabolisms in the liver. It inhibits hepatic glucose production via the IRS-Akt-FoxO and IRS-aPKC-CREB pathways[22, 23], and promotes de novo lipogenesis via Akt and aPKC mediated activation of Srebp-1c, although detailed mechanisms remain unclear[24–26]. The observed phenomenon of hepatic selective insulin resistance, namely that hepatic glucose production becomes resistant to insulin while de novo lipogenesis remains unabated or even gets enhanced, is a long-standing paradox in T2D[27]. Although, recent results point to a more prominent role of altered nutrient delivery to the liver[28], defects in insulin signaling might also contribute to the development of hepatic selective insulin resistance[29–32].

We integrated current experimental knowledge of the proximal hepatic insulin signaling network into a mathematical model. The response of the model to physiological insulin profiles is investigated. In particular, the model is quantitatively informed by two in vivo data sets from rodent studies, where hepatic insulin signaling was measured after refeeding[33] or after various patterns (pulsatile/constant/T2D) of pre-hepatic insulin infusion[18]. The model analysis explores the information hidden in these two data sets and reveals distinct characteristic features of the hepatic insulin signaling network sufficient to understand the dynamic features of insulin secretion, their impact on hepatic insulin signaling and the emergence of selective hepatic insulin resistance.

**Table 1 Summary of the tested models**

|  | # fitted para | Best RMS | Best AICc | Best mAICc | Note |
|---|---|---|---|---|---|
| M0 | 37 | 2.00 | −16.68 | −16.37 | Full model |
| M1 | 35 | 2.02 | −18.35 | −17.93 | Based on M0; feedback from aPKC on insulin receptor internalization removed |
| M2 | 33 | 1.75 | −21.46 | −19.73 | Based on M1; negative feedback from Akt on IRS1 removed |
| M2a | 33 | 2.30 | −19.21 | −19.55 | Parallel to M2; Akt auto-feedback replaced the feedback from Akt to insulin receptor |
| M3 | 31 | 2.16 | −22.46 | −21.55 | Based on M2; negative crosstalk from aPKC on Akt removed |
| M4 | 29 | 2.53 | −24.0 | −22.76 | Based on M3; positive feedback from Akt on insulin receptor removed; minimal model |
| M5 | 35 | 2.73 | −14.95 | 16.64 | Based on M0; negative feedback from aPKC on IRS1 removed |
| M6 | 35 | 3.22 | −12.02 | −6.97 | Based on M0; auto-phosphorylation of aPKC removed |

The level of model complexity decreases from M0 (full model) to M4 (minimal model). Removing from the full model either of the two feedbacks in the minimal model (M5 and M6) leads to fitting failure. RMS is the root mean square difference between model simulation and measured data. Please note that for both RMS and AICc, smaller values mean better performance

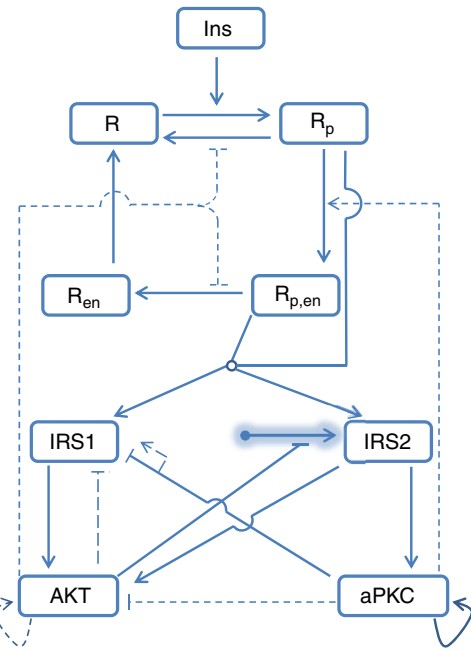

**Fig. 1** Scheme of the mathematical model. Insulin (Ins) induces insulin receptor (R) phosphorylation (R$_p$), which then undergoes internalization (R$_{p,en}$), dephosphorylation (R$_{en}$) and reinsertion into cell membrane (R). Phosphorylated insulin receptors (both R$_p$ and R$_{p,en}$) activate IRS1 and IRS2, which both activate Akt. Only IRS2 activates aPKC. Synthesis of IRS2 (shadowed arrow with round end) is subject to inhibition induced by AKT. Both, AKT and aPKC have positive auto-feedback. aPKC inhibits IRS1 and AKT activity. Broken lines indicate feedbacks that have been suggested in the literature but are dispensable for data fitting. Shadowed arrows with round ends indicate that the synthesis and degradation of the pointed element are considered in the model. The open circle at the place of merging lines indicates shared downstream targets

## Results

**A mathematical model of hepatic insulin signaling.** Many feedbacks/crosstalks have been reported to shape insulin signaling and play a role in the development of hepatic insulin resistance. For example, mTORC1, and its effector kinase S6K, limit insulin signaling by threonine/serine phosphorylating IRS1. This negative feedback has been implicated in the development of insulin resistance in several animal models[34]. Here, we use data derived from animals without insulin resistance or other type of metabolic dysfunctions. The data are limited to 2 h after refeeding or 30 min after insulin infusion. It is not clear to which extent, if any, each reported feedback/crosstalk plays a role in this limited time window. To explore the role of various feedbacks/crosstalks, we: (1) developed a series of mathematical models with different feedback structures; (2) fitted each model to the two data sets; (3) compared model performance in fitting by using the Akaike information criterion with correction (AICc)[35], which evaluates model quality based on both quality of fitting and the number of free parameters; and (4) compared the robustness of the response of the fitted models to variations in the input, i.e., insulin level, by the mean of AICc, (mAICc, see Supplementary Methods). It is important to ensure that the response of the model is robust to small variations in insulin level, since such a property is common to many physiological systems. As the pre-hepatic insulin levels in both, the refeeding experiment and the infusion experiments, were not measured, the requirement of a robust response could compensate potential errors associated with estimated insulin levels (see Supplementary Methods).

We considered the following feedback/crosstalk mechanisms:

1. Feedback from aPKC on insulin receptor internalization[36].
2. Feedback from Akt on insulin receptor de-phosphorylation[37], and, optionally, auto-feedback of Akt[38], whose effect partially overlaps the feedback from Akt on insulin receptor de-phosphorylation.
3. Feedbacks from Akt and aPKC on IRS1 activity through various serine/threonine phosphorylation[39]. Serine/threonine phosphorylation of IRS1, for example of Serine-302, can have either a positive or a negative effect on IRS1 activation, depending on the phosphorylation status of the other sites[40]. Since the feedback from Akt on insulin receptor de-phosphorylation[37] already presents a positive feedback on IRSs, we tested only the negative feedback from Akt on IRS1. Both positive and negative feedbacks from aPKC on IRS1 have been tested and it turned out that only the negative feedback is consistent with the data.
4. Transcriptional inhibition of IRS2 by Akt, which we kept active because of direct experimental evidence, see Fig. 4b in ref. [33].
5. Akt inhibition by aPKC[41, 42].

The mathematical models were validated by simultaneously fitting two published data sets. The refeeding data set was derived from a mouse study[33], where hepatic phosphorylation of IRS1, IRS2, and Akt were measured for a period of 6 h after feeding. Since indirect systemic effects onto the liver were reported 3 h after feeding[43], we used the data from the first 2 h only. The time course of pre-hepatic insulin was derived from the serum levels, which peaked at 30 min after feeding[33].

Second, an infusion data set from a study with rats was used[18]. Endogenous insulin was inhibited and exogenous insulin injected directly into the portal vein. Insulin was provided either in pulses (high dose mimicking normal and low dose mimicking a T2D condition) or constant (high dose). Considering that the expected insulin profile might be established only after some transient flow instability (see Supplementary Fig. 1 in the supplement of ref. [18]), we ignored the data obtained before 10 min and relied on the Akt phosphorylation levels at 30 min.

We carried out numerical optimization tasks for all the models (Table 1, Fig. 1 and Supplementary Methods). We identified a minimal model with 29 parameters, including 25 parameters in the model and 4 scaling factors, (M4 in Table 1; Fig. 1 solid lines) that keeps only three feedbacks: (1) aPKC inhibiting IRS1; (2) aPKC auto-phosphorylation; and (3) Akt suppressing IRS2 transcription. The importance of the former two feedbacks was confirmed by additional optimization studies. Removing any of those from the model led to fitting failure (see M5 and M6 in Table 1).

We further carried out parameter identifiability analysis (Methods section and Supplementary Methods), which showed that only 14 parameters are well confined (Supplementary Fig. 1). The 13 solutions (parameter sets) resulting from the identifiability analysis are included in the Supplementary Data 1, and the associated fitting results are shown in Supplementary Fig. 2. The following results in this paper (Figs. 2–6) are based on the best fitting result of the minimal model (M4) and the model predictions hold true for all 13 solutions. As the difference between mAICc of M3 and M4 is less than 2 units, the consistency of all findings was confirmed in model M3 (Supplementary Fig. 3).

**aPKC behaves as a bimodal switch.** The refeeding data set (Fig. 2, column Feeding) is well reproduced by the model. The model predicts that the dynamics of activation of the involved

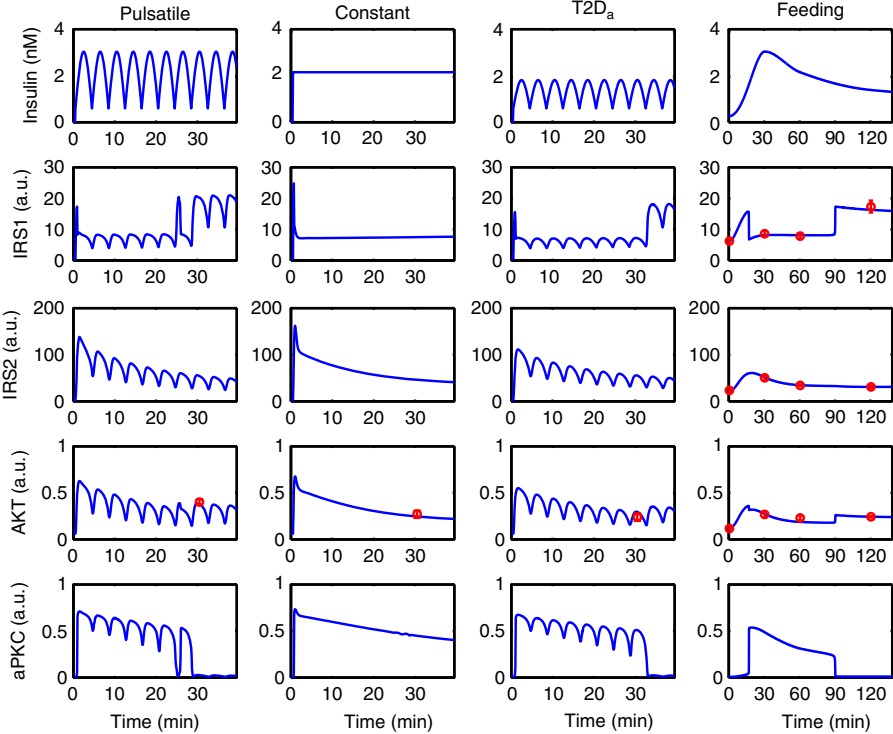

**Fig. 2** Best fitting result of the minimal model (M4 in Table 1; solid arrows in Fig. 1). The activities of the signaling molecules are plotted with arbitrary unit (a.u.). Each column corresponds to one specific experiment. Experimental data are indicated by red circles with error bars (s.e.m). See Supplementary Fig. 2 for similar results based on other parameter sets

players exhibit non-linearities: IRS1 and Akt (Fig. 2, column Feeding) showed a sharp drop 10–15 min after the beginning of feeding, followed by a rebound at about 90 min. The drop and the rebound of IRS1 and Akt were associated with a switch-like activation and deactivation of aPKC.

These findings were confirmed for the infusion data set. The specific phosphorylation levels of Akt in the three infusion modes were also associated with different states of aPKC deactivation. In the pulsatile case (Fig. 2, column Pulsatile), aPKC was fully deactivated around 30 min after infusion and a rebound of IRS1 and AKT activity occurred at the same time. However, in the case of constant infusion of the same amount of insulin (Fig. 2, column Constant; see also Supplementary Fig. 2), aPKC still showed some level of activity after 30 min, which was associated with a lower activation level of Akt and IRS1. Notably, while the association of IRS1 and Akt activity with a switch in aPKC is consistent with all experimental data, a constant infusion of insulin failed to switch-off aPKC activity completely.

Next, we explored the reason behind the differential aPKC behavior in response to the pulsatile and constant infusion mode. A bifurcation analysis of the aPKC sub-system revealed that aPKC undergoes a bistable switch under the control of activated IRS2 (Fig. 3). The bistable switch of aPKC gives rise to hysteresis: the threshold level of active IRS2 for aPKC switch-on (LP1 in Fig. 3), is higher than that for aPKC switch-off (LP2 in Fig. 3). Due to the transcriptional inhibition by Akt, IRS2 activity approached a steady-state after an overshoot, in the case of constant infusion (Fig. 2, column Constant; the numerical value is ~44 a.u.). The steady-state level of IRS2 was higher than LP2, which resulted in the aPKC steady-state level remaining on the upper branch of the bifurcation structure. However, for the case of pulsatile infusion, IRS2 approached the steady-state with a damped oscillation (Fig. 2, column Pulsatile). Due to the fluctuations between the peak and the trough of the oscillation

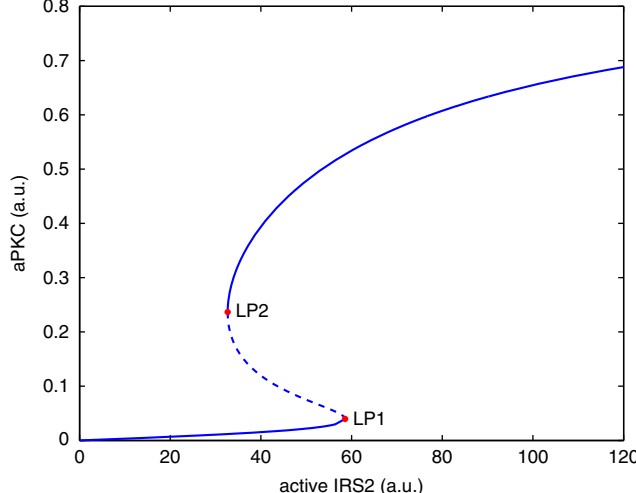

**Fig. 3** Bifurcation diagram of the aPKC sub-system with the level of active IRS2 as control parameter. The broken line between the two marked limit points (LP1 and LP2) denotes unstable steady states

(numerical values between 54 and 28 a.u. around 30 min), active IRS2 could cross the threshold LP2 and switch-off aPKC.

**The timing of aPKC switch-off reflects the nutritional state.** Starting from the best fit in Fig. 2, we scaled the estimated insulin level to 85% (Fig. 4 black curves) and 115% (Fig. 4 blue curves) and investigated the corresponding responses. The following results persist for other parameter combinations (Supplementary Fig. 4).

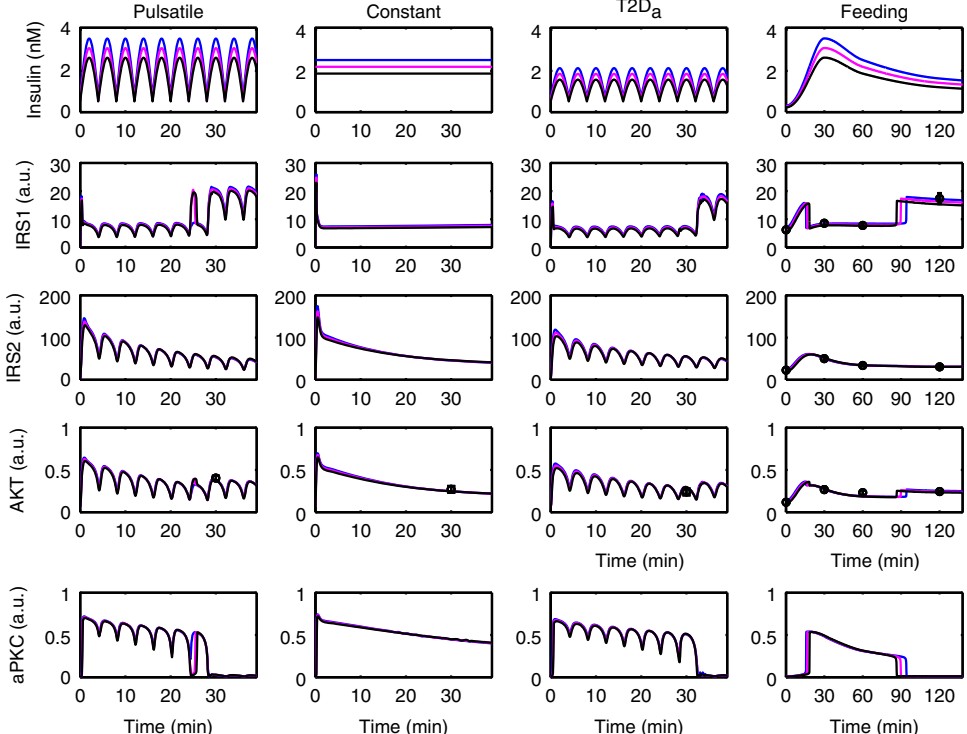

**Fig. 4** Response of the model to three doses of insulin. The activities of the signaling molecules are plotted with arbitrary unit (a.u.). Each column corresponds to one specific experimental set-up (see text). The best fit in Fig. 2 is shown in magenta which defines a reference amount of insulin. Using 85 and 115% of this insulin is shown in black and blue, respectively. The timing of aPKC switch-off and IRS1-AKT rebound is sensitive to the insulin level, while the activation level is not. See Supplementary Fig. 4 for similar results based on other parameter sets

All the simulated curves matched the data points (Fig. 4). This indicated that the insulin levels investigated were approaching a saturation level, which is very likely to be true in the postprandial case[3]. However, the simulation showed that, in the refeeding experiment, the switch-off of aPKC and the rebound of IRS1 and Akt were delayed and advanced in the case of higher and lower insulin dose, respectively. The levels of aPKC, IRS1, and Akt before and after their state transitions were nearly independent of the insulin level. This suggests that the information encoded in insulin doses, and therefore the information on the nutritional state sensed by pancreatic β cells, is in part decoded by the timing of hepatic IRS1, Akt, and aPKC state transitions rather than in the activation level of these molecules. It is tempting to hypothesize that the timing of IRS1, Akt, and aPKC state transitions has a functional importance in the hepatic insulin signaling network.

**aPKC onset and switch-off depend on insulin dynamics**. Considering that impaired first-phase insulin in (pre-)diabetes is associated with impaired glucose tolerance, we investigated the effects of different dynamic features of first-phase insulin, including the peak level and the slope of the increase (Fig. 5a). We constructed a fast rising insulin (Fig. 5, blue), denoted as "normal" in the following. Deviations from the normal insulin dynamics include an impaired slope with a normal peak (Fig. 5, cyan), as well as an impaired slope and peak, which was designed to mimic T2D (Fig. 5, magenta, labeled as T2D_b). The three insulin profiles considered have the same area under curve (AUC) (Fig. 5f); however, with different dynamic features (Fig. 5g, h). In the model, variations in the dynamic features of the insulin profile resulted in a changed timing of aPKC switch-on and off (Fig. 5e). The drop and rebound of IRS1 and Akt were shifted correspondingly (Fig. 5b, d). Changes in the first-phase insulin also

changed the peak level of IRS2 and aPKC (Fig. 5c, e), but not that of IRS1 and Akt (Fig. 5b, d), suggesting that IRS2 and aPKC are more sensitive to first-phase insulin than IRS1 and Akt. Although the peak of IRS2 showed a more sensitive response to first-phase insulin, the overall activity of IRS1, as quantified by the AUC, was more sensitive to different insulin profiles (Fig. 5f), which could be explained by the aPKC activity (Fig. 5f).

Following the reasoning that the observed abrupt transitions in the activity state of these molecules have a critical functional importance, we investigated the signaling before and after the switch-off of aPKC in two windows: 0−1 and 1−2 h, and calculated the AUC in each window separately. In the first hour (Fig. 5g), IRS2 and aPKC followed the trend of insulin while IRS1 and Akt were barely sensitive to insulin patterns. In the second hour (Fig. 5h), however, IRS2 and aPKC still followed the trend of insulin while IRS1 and Akt showed the opposite trend (Fig. 5h). Similar results were found for other parameter sets (Supplementary Fig. 5). While the network (Fig. 1) is widely interconnected, this finding suggests a functional separation of IRS1 and IRS2 in the course of hepatic insulin signaling.

Since impaired insulin pulses are also associated with impaired glucose tolerance in T2D, we investigated the effect of the amplitude of insulin pulses on hepatic insulin signaling in the model. Obviously, in vivo pre-hepatic pulsatile insulin shows more stochasticity than what we constructed here with a sinusoid function (Supplementary Methods). However, we still can learn basic principles of pulsatile insulin signaling with the mathematical model. We constructed two pulsatile insulin profiles on the basis of "normal" first-phase insulin (blue curve in Fig. 5a) with high and low pulse amplitudes (Fig. 6). The amplitude of the latter is one quarter of the former but with the same amount of insulin (Supplementary Fig. 6). Both profiles induced an aPKC switch-off within an hour after the switch-on, which is earlier

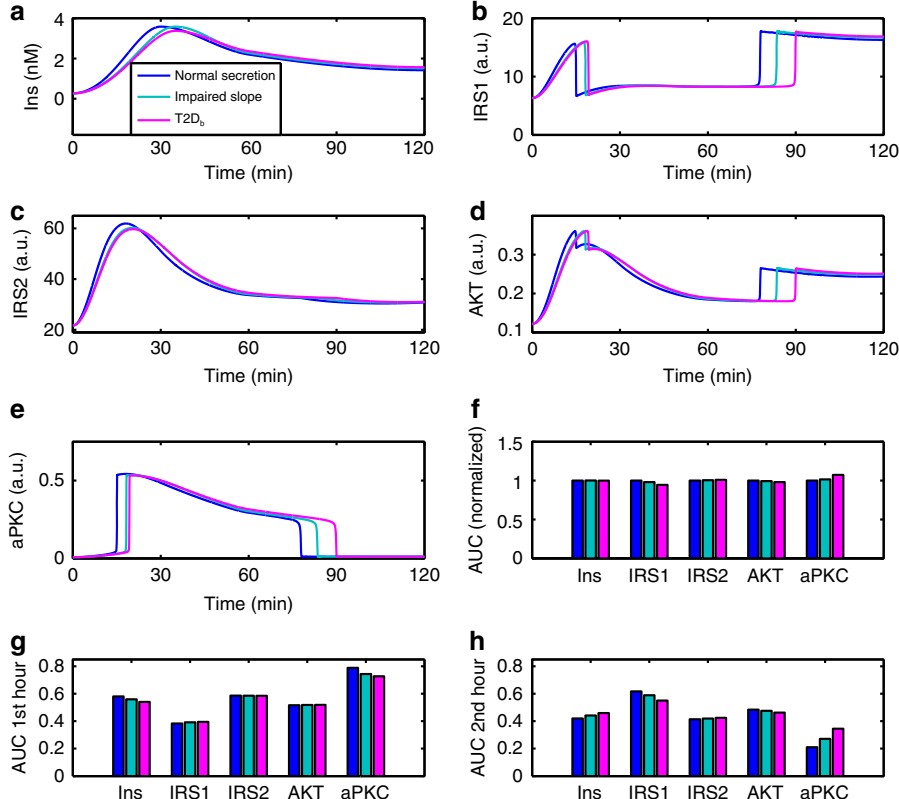

**Fig. 5** Response of the model to insulin profiles with different dynamic features. Impaired first-phase insulin (cyan) lead to a delayed aPKC switch-off and a delayed IRS1 and AKT rebound. Reduced insulin peak levels (magenta) further delay these activity transitions. See Supplementary Fig. 5 for similar results based on other parameter sets

than with the non-pulsatile insulin (80 min, Fig. 5). Furthermore, a higher insulin amplitude is associated with an earlier aPKC switch-off (Fig. 6). The rebound time point of IRS1 and Akt was shifted correspondingly. Thus, in the model, the effect of reduced insulin pulse amplitude is similar to the one observed in response to impaired first-phase insulin (Supplementary Fig. 6).

## Discussion

The first comprehensive mathematical model of the insulin signaling network appeared 15 years ago[44]—here referred to as the Sedaghat model, which, for many years, constituted the basis for subsequent mathematical models of insulin signaling. The Sedaghat model was mostly based on in vitro experiments of adipocytes. Insulin signaling in hepatocytes differs from that in adipocytes in several aspects, including the dominant isoforms of IRSs[45], the role of IRS1 in Akt and aPKC activation[46] and in vitro responsiveness to insulin[31]. As our aim was to investigate the importance of particular in vivo insulin profiles in hepatic signaling, and considering that the two in vivo data sets used here contain only limited numbers of time points, we adopted a parsimonious strategy in designing our model (Supplementary Methods). A subset of parameters was non-identifiable, which prohibits a quantitative interpretation of these parameters. In addition, there is no guarantee that the entire parameter space has been searched and the found solutions are likely local minima. Nevertheless, the model predictions are qualitatively consistent over all the solutions found by the parameter identifiability analysis, such that we consider them as reliable hypotheses, which have to be validated in further experiments. Our modeling study showed that a single mechanism, namely an IRS2-dependent bistable switch in aPKC, is the key mechanism to explain the two

data sets. Investigation of model responses to the in vivo insulin time courses revealed that the timing of aPKC suppression is delayed by increased insulin doses while the activation levels of the associated signaling molecules remained unaltered. The model further suggests that the timing of aPKC suppression is sensitive to dynamic features of pre-hepatic insulin: an impaired first-phase insulin as well as a reduced amplitude of insulin pulses resulted in delayed aPKC suppression. The beauty of this simple explanation of two complex data sets is that it also explains how selective insulin resistance would develop in a healthy (insulin sensitive) liver, when exposed to diabeticinsulin secretion, as will be further elaborated below.

The mechanisms underlying integrated control of hepatic glucose production and de novo lipogenesis by insulin are far from being clear. There is increasing evidence that the sequence of events, in particular FoxO inhibition and Srebp-1c activation, is important[29]. Our findings suggest a functional importance for aPKC suppression, which indicates a low level of IRS2, and consequently, FoxO1 activity, in the determination of the sequence of events in the insulin signaling network. The two processes, hepatic glucose production and de novo lipogenesis, seem to be mutually exclusive on the transcriptional level[47, 48]. Here, with the help of the mathematical model, we discuss the complex interactions between the two important kinases involved in hepatic insulin signaling, namely Akt and aPKC, and show how these are involved in selective hepatic insulin resistance in response to impaired insulin secretion.

Both Akt and aPKC contribute to the inhibition of hepatic glucose production (via Foxo1 and CREB-CBP-CRTC2 inhibition, respectively), and promotion of de novo lipogenesis in the liver (largely via Srebp-1c induction)[22–26]. However, they also compete with each other. Akt exerts transcriptional inhibition on

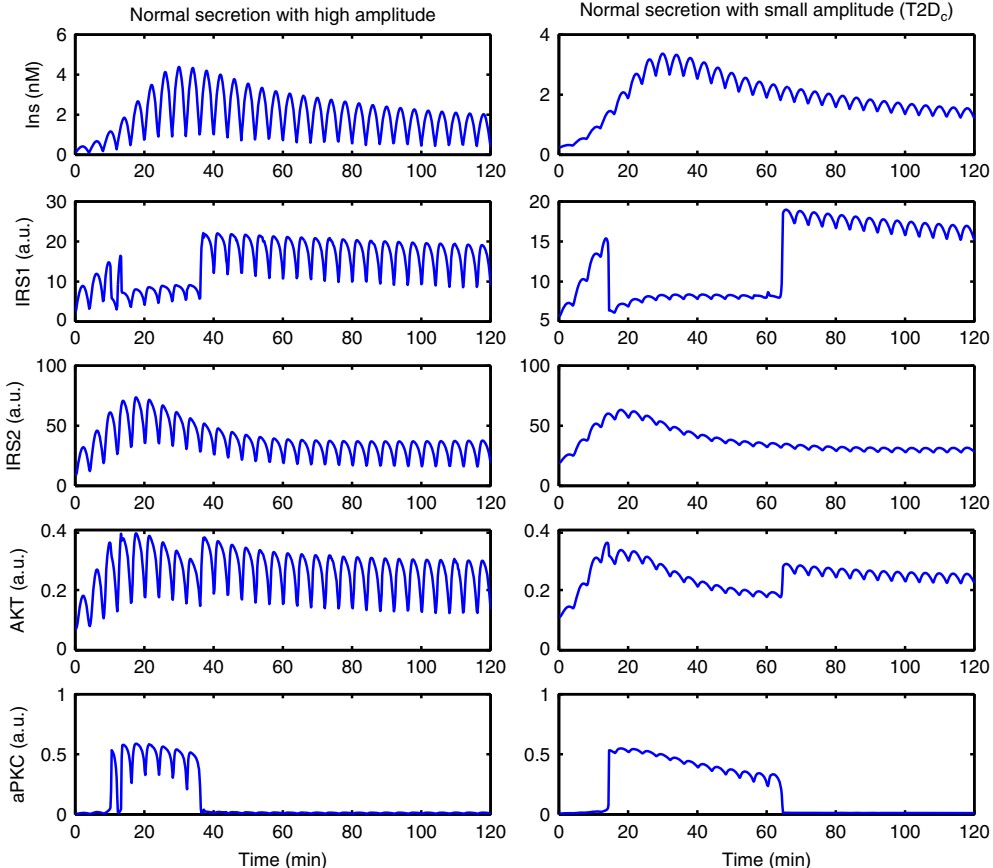

**Fig. 6** Responses of the minimal model to insulin with high (left) and low (right) pulse amplitude. The amounts of insulin in the two cases (quantified by AUC) are equal. See Supplementary Fig. 6 for similar results based on other parameter sets

IRS2. Importantly, aPKC is activated only by IRS2[21]. aPKC inhibits, by direct phosphorylation, the activity of IRS1 and Akt[41, 42, 49]. Unlike aPKC, Akt is activated by both IRS1 and IRS2. Thus, Akt and aPKC form a mutually inhibitory crosstalk, albeit on different time scales: aPKC inhibits IRS1-Akt instantly while Akt inhibits IRS2-aPKC with a delay. The mathematical model suggests that the mutually inhibitory crosstalk between Akt and aPKC, in particular the differential kinetics of the inhibitions, is the key mechanism for explaining the two data sets, as indicated by the structure of the minimal model (Fig. 1 solid lines).

The mathematical model predicts that aPKC is activated by IRS2 via a bistable switch. This was derived from two sets of in vivo measurements, in which aPKC itself was not determined. Biochemical studies suggested that full activation of aPKC requires (auto-)trans-regulation[50, 51] followed by a conformational change that leads to the release of pseudo-substrate auto-inhibition. Further experimental studies are required to prove the existence of a postprandial bistable switch of aPKC in the liver. As hepatic aPKC can also be activated by ceramide, which is relevant in diabetes development, the prediction of the presence of an insulin induced bistable aPKC switch should be first tested in situations where ceramide levels are low.

Currently there exist controversial experimental results concerning insulin induced aPKC activation and its role in effecting postprandial Srebp-1c induction. Insulin activated aPKC has been shown to be required for chromatic remodeling favorable for transcription of the lipogenic genes[52]. In addition, aPKC phosphorylates CREB binding protein (CBP), which then can acetylate Srebp-1c and enhance the binding of Srebp-1c to target lipogenic promoters[53, 54]. The importance of aPKC in hepatic lipogenesis is further supported by the effects of aPKC inhibitors[55] and loss-of-

function studies[24, 56]. While a growing body of evidence supports the role of aPKC in mediating insulin's effect on Srebp-1c transcription after feeding, Wan et al.[57] could not observe increases in hepatic aPKC phosphorylation following insulin injection. Our mathematical model predicts that aPKC activity undergoes fast switch-on and switch-off and is limited in a time window (from 15 to 90 min after refeeding), providing a potential explanation for the conflicting observations of hepatic aPKC activation.

Simulations with different insulin profiles showed that the timing of switch-on and the peak level of aPKC are sensitive to the slope of first-phase insulin, but not the peak level of first-phase insulin (Fig. 5). This was rooted in the upstream activator IRS2, which, due to the transcriptional inhibition from Akt, peaked before 30 min (Fig. 5c). In other words, IRS2 and aPKC reached their peak levels earlier than the first-phase insulin did. Unlike the timing of switch-on, which showed only minor differences in response to different insulin profiles, the timing of aPKC switch-off showed more prominent differences in response to different insulin profiles (Figs. 5, 6) and different insulin doses (Fig. 4). This again was rooted in IRS2, whose post-peak activity became quite flat (Figs. 4, 5 in this study, and Fig. 5d in ref. [33] showing IRS2 activity until 6 h after feeding). The post-peak IRS2 activity decreased towards the threshold for aPKC switch-off (LP2 in Fig. 3) at a slow pace, thus giving rise to the prominent difference in the timing of aPKC switch-off. These results suggest that the features of IRS2 kinetics, which reflect the level of FoxO1 suppression, are critical in shaping the response of the insulin network. In T2D patients, the presence of the second-phase hyperinsulinemia, together with the flat post-peak activity of IRS2, contributes to the hyperactivity of aPKC in the liver of these patients (Figs. 5, 6).

The infusion data set demonstrated that the liver is more sensitive to pulsatile insulin delivery pattern in terms of signaling, as compared to constant delivery or to a pattern mimicking T2D[18]. Please note that the T2D$_b$ insulin profile in Fig. 5 differs from T2D$_a$ in Figs. 2, 4. The former mimics impaired first-phase insulin while the latter mimics impaired pulsatile insulin. Our simulation results with the more realistic insulin profile (pulsatile + first-phase, T2D$_c$ in Fig. 6) suggests that it renders the control of IRS1 and Akt rebound and of aPKC switch-off more robust, since the fluctuation between the peak and the trough of the pulses allows an earlier crossing of the threshold for aPKC switch-off (LP2 in Fig. 3) than non-pulsatile insulin. However, since in vivo prehepatic insulin is more stochastic than what we constructed here with sinusoid functions, more studies are required to fully understand the advantages associated with pulsatile insulin. For example, Satin et al.[58] have studied the response of the Sedaghat model to constant and pulsatile insulin and proposed that the negative feedbacks targeting IRS molecules decay during the interval between two consecutive pulses and therefore allow a higher Akt response to insulin pulses. This proposal is consistent with a theoretical analysis, which attributes the enhanced response to pulsatile hormone to the sigmoidal nonlinearity in the dose-response curve resulting from negative feedbacks in the system[59]. The mechanism proposed in our model relies on the bistable switch of aPKC, whose dose-response curve (Fig. 3) is twisted more than a sigmoidal function. An apparent difference is that the enhanced response induced by sigmoidal nonlinearity occurs during each pulse of insulin while the enhanced response by bistability occurs only once, i.e., when aPKC is switched off. It is likely that both mechanisms are involved in hepatic insulin signaling and might be dominant in different contexts, like fasting and feeding. In addition, the feedbacks that were determined dispensable for fitting in this paper might be able to fine-tune the behavior of the network in a context specific manner. Moreover, considering that both aPKC and Akt play a role in controlling insulin receptor internalization[36] and that an internalized insulin receptor is associated with transient higher exogenous tyrosine kinase activity[60], the proposal that insulin-pulse-entrained receptor recycling contributes to the enhanced effects of pulsatile insulin deserves more detailed investigation. Further, it has been shown that in T2D, and even prediabetes, not only the ratio between insulin and glucagon is dysregulated[61], but also the anti-phase relationship between pulsatile insulin and glucagon secretion is impaired/lost[62]. The glucagon-PKA pathway interferes with many molecules involved in insulin signaling, for example Akt[38]. It is necessary to bring the glucagon pathway into consideration, in order to understand selective insulin resistance in the context of prediabetes progression.

Our model showed that, due to the aPKC switch, the activities of IRS1 and IRS2 are mutually exclusive after feeding, except for the first 15 min. Thus, the effect of insulin is separated into two time windows: an IRS2 window starting from feeding lasting until 1−2 h after feeding, and an IRS1 window starting from 1−2 h after feeding. Given the structural difference between IRS1 and IRS2, it has been hypothesized that IRS1 can mediate insulin dependent mitogenic and metabolic effects via its ability to activate both the ERK and the PI3K pathway, while IRS2 can only mediate insulin metabolic effects[63]. Here, we further hypothesize that the first IRS2 window would be mostly devoted to hepatic glucose production control and the second IRS1 window more to de novo lipogenesis and mitogenic control. In this scenario, a delayed aPKC switch-off interferes with the transition from hepatic glucose production inhibition to de novo lipogenesis activation. In a healthy setting, hepatic glucose production and de novo lipogenesis are well separated in time, indicated by a prompt switch-off of aPKC. However, in a pathological state, due to delayed switch-off of aPKC, hepatic glucose production and de novo lipogenesis become overlapping. Our simulations showed that impaired insulin secretion and the subsequent delayed aPKC switch-off lead to (1) less aPKC activity in the first IRS2-associated hepatic glucose production inhibition window, and (2) more aPKC activity in the second IRS1-associated de novo lipogenesis activation window.

Importantly, Akt activity is far less sensitive to the dynamic features of insulin than aPKC, in the current context of an insulin sensitive liver (Fig. 5, Supplementary Fig. 5). This condition is different from insulin resistance in diabetes for which Akt activation by insulin is impaired due to other mechanisms[64]. This diabetic-insulin induced aPKC activity pattern in "insulin sensitive" liver correlates well with the phenomenon of selective hepatic insulin resistance, given the documented ability of aPKC to inhibit gluconeogenesis[22, 65], promote lipogenesis and activate NF-κB[51]. This idea is further supported by the following facts: (1) postprandial appearance of endogenous glucose reaches its minimal level before 1 h after feeding[3]; (2) Srebp-1c expression requires hours[66]; and (3) hepatic ERK is hyperactive in diabetic mouse models[67]. While there is experimental evidence for our new model of selective insulin resistance, a mathematical analysis of quantitative data was required to reveal the implications of the data for our understanding of insulin resistance. Recent experimental advances pointed to the important roles of certain distal part of the insulin signaling network in inducing selective insulin resistance, such as mTORC1[30], mTORC2[32], and FoxO[68]. Once data suitable for model development become available, it would be interesting to extend the hepatic insulin signaling model to include these molecules and to address selective insulin resistance in a more complete signaling network and in various contexts.

## Methods

**Model and simulation.** The mathematical model of hepatic insulin signaling (Fig. 1) describes the following processes: (1) insulin receptor (R) activation, deactivation, internalization, and reinsertion; (2) IRS1 activation and deactivation; (3) IRS2 synthesis, degradation, activation, and deactivation; (4) AKT activation and deactivation; and (5) aPKC activation and deactivation. All the simulation work was based on the SBPD toolbox for Matlab[69]. A differential evolution based global optimizer was employed to fit the parameters in the model. The bifurcation analysis utilized Matcont[70].

**Parameter identifiability and prediction reliability.** The biological system under consideration requires a minimum complexity of the model in order to capture the experimental observations. In view of the available experimental data, a subset of the model parameters is non-identifiable. Non-identifiability of some parameters prohibits a quantitative interpretation of those parameters but does not imply that the underlying model structure is fallacious. However, the parameter identifiability analysis generates a large number of solutions which would lead to different model behavior. As a consequence, we face the problem of deciding which solution describes the nature best. In order to generate reliable model predictions, a strategy is required that allows choosing model solutions with additional criteria. We took the following five-step procedure in order to generate reliable model predictions.

1. Determine many possible solutions with a genetic fitting algorithm based on the resulting RMS-value. Especially, in order to generate solutions as diverse as possible, parallel fittings were carried out where one particular parameter is confined in different regions. This was done for every parameter for which relevant experimental data are not available.
2. Test all solutions from step 1 against the additional experimental constraint, which states that the recycling of insulin receptor can follow physiological insulin pulses quickly. Inconsistent solutions (see examples in Supplementary Fig. 7) were excluded for further analysis.
3. A minimum distance criterion was applied to solutions from step 2, in order to ensure the independence of the solutions, since solutions from the same local minimum might exist.
4. The solutions from step 3 were filtered by the mAICc criterion, which ensures a robust model behavior against variations of insulin levels.
5. Derive a set of predictions and keep only those predictions that appear consistent across all the solutions resulting from step 4.

This procedure should generate reliable model predictions despite non-identifiability of a subset of parameters and despite the fact that we might have

missed the global minimum in the parameter space. However, the derived predictions have still to be validated by additional experiments.

**Data and code availability**. The authors declare that all data supporting the findings of this study are available within the paper and its Supplementary Information files.

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

## Acknowledgements

This work was supported by the Helmholtz Association cross-program activity "Metabolic Dysfunction and Human Disease". M.M.-H. was supported by the Helmholtz Initiative on Personalized Medicine—iMed. We thank Dr Sebastian Binder for revising the manuscript and valuable comments.

## Author contributions

G.Z. designed the study, carried out the research, and interpreted the results. M.M.-H. supervised the study. G.Z. and M.M.-H. wrote the manuscript. D.W. and I.S. assisted in study design and reviewed the manuscript.

## Additional information

**Competing interests:** The authors declare no competing financial interests.

