## [Peer Review file · Nature Communications]

Reviewers' comments:

Reviewer #1 (Remarks to the Author):

The authors analyzed the past literature data using mathematical method. Under physiological conditions, aPKC was activated by increase of insulin along with Irs2 activation, leading to decreased activation of both IRS-1 and Akt. When activation of aPKC was switch-off, IRS-1 and Akt activation was induced simultaneously. In pathophysiological conditions, when de-activation of aPKC was delayed due to impaired insulin slope and non-pulsatile insulin, activation of IRS-1 and Akt was also delayed. These findings suggest that delayed Irs2-aPKC pathway may cause prolonged gluconeogenesis, and both gluconeogenesis and lipogenesis may occur simultaneously in type 2 diabetes.

Although authors investigated using a mathematical model, causal relationship remains unclear. However, this study is of considerable interest. Several points need to be clarified as follows:

1. The importance of mTORC1 and FoxO1 has been pointed out as a mechanism of selective insulin resistance (Proc. Natl. Acad. Sci. U. S. A. 107, 2010, 3441-3446; Cell Metab 23, 1154-1166, 2016). Are the authors able to investigate activation of mTORC1 or FoxO1 using the same mathematical model?
2. Is expression levels of gluconeogenesis-associated genes such as PEPCCK and G6Pase regulated by Irs2-aPKC pathway?
3. Activation of IRS-1 and Akt seemed to be similar degree between pulsatile injection and T2D injection (Fig.2, Fig.4). However, in type 2 diabetes, lipogenesis was enhanced. Please explain this discrepancy in discussion.

Reviewer #2 (Remarks to the Author):

The authors developed a mathematical model of the early cascade of hepatic insulin signalling. Two experimental data sets in rats allowed tuning the model, which is then used to simulate the time course of measured and unmeasured variables in response to different pattern of insulin stimulation. Based on model predictions, the most relevant feedback/crosstalk mechanisms among those reported in the literature are identified, and interesting hypothesis on the mechanisms underlying the more efficacious hepatic insulin response to pulsatile insulin delivery are formulated. The paper is well organized and clearly written, but some aspects of the modeling process appear to be weak, or at least are not clearly explained/motivated. This is particularly important in the present study given the limited amount of available data which strongly affect the reliability of quantitative model predictions.

Major

1. The model is not discussed in the context of previous literature, where several models of insulin signalling have been published. I invite the authors to compare their model with state-of-the-art-models, and to elaborate on the rationale underlying the choice of the state variables they included in their model.

I also point out to the authors' attention the recent paper by Satin et al, Mol Aspects Med 2016, reporting simulations of a key model of insulin receptor signaling during continuous and pulsatile insulin stimulations.

2. The proposed model has 39 parameters, to be quantified based on 15 samples, related to 4 different patterns of insulin stimulation. As the authors acknowledged, the model is obviously not

uniquely identifiable, thus in general terms the reliability of parameter estimates is questionable, as well as the predictions derived from them. This raises doubts on the biological hypothesis based on model predictions, particularly of aPKC profiles, which are not supported by any data at all.

3. By using a genetic algorithm, 50 points in the 39-dimension parameters space able to provide a reasonable fit were identified, but the information on how these points were selected is missing. No information is given on how the errors affecting the measurements are handled. I thus invite the authors to explain in the method section the identification procedure they adopted as well as the criteria they used to assess the validity of the estimates. A mandatory criterion is the biological plausibility of estimated values, in light of representative values which are available in the literature for most parameters.

4. In the present paper, only the 30 min AKT data from the study by Matveyenko et al. are used in model fitting, while the early data on IRS1 and IRS2 activation, available from the same study, are excluded. The authors justify their choice, writing that: "expected insulin profile might be established after some transient flow instability" which is rather cryptic to me. Given that the receptor activation is very fast, and that the model considers the early cascade, information on the dynamics of the involved substrates can be obtained only by early samples. Luckily, they are available in the Matveyenko et al. study, so why don't use them?

5. Based on model predictions, the " feedback/crosstalk mechanisms that are indispensable for the model to fit the data" are selected. Can the authors please explain the criteria they implement to compare the alternative model structures, and to select the one they call the minimal one?

Minor

1. References in the Supplementary Information are labelled with numbers, and it is not clear to which item they refer to.

2. Figure S2 shows distribution (ranges?) of fitted parameters, but labels do not always correspond to the symbols used in Model section of Supplementary Information. Please check.

3. If I understand correctly, the boxes in figure S2 represent the mean +/- SD intervals. Thus I'd expect they are symmetrical with respect to the mean, which is not the case for a few of them (e.g. e0p1e1). Please explain.

4. I found some difficulties in reading the model equations in the Supplementary Information. Introducing some additional symbols in Figure 1, and the definition of symbols before the equations (and not at the end of the paragraph) could help the reader.

5. The pulsatile insulin input is modeled by a sinusoidal function, which is an approximation of the input used in the Matveyenko et al. study. The approximation is likely to introduce minor approximations on model simulations, but I don't see the rationale for not using the real profile.

Thanks to the critical comments from the two reviewers, the manuscript is now improved in many aspects, including a more thorough explanation of the modeling process, a more extensive discussion in relation to published results and a more compact model formulation. The model predictions turned out robust against the different model variants we now tested.

Below we answer point by point to the reviewer's comments, which are repeated below in italic font.

Reviewer #1 (Remarks to the Author):

The authors analyzed the past literature data using mathematical method. Under physiological conditions, aPKC was activated by increase of insulin along with Irs2 activation, leading to decreased activation of both IRS-1 and Akt. When activation of aPKC was switch-off, IRS-1 and Akt activation was induced simultaneously. In pathophysiological conditions, when de-activation of aPKC was delayed due to impaired insulin slope and non-pulsatile insulin, activation of IRS-1 and Akt was also delayed. These findings suggest that delayed Irs2-aPKC pathway may cause prolonged gluconeogenesis, and both gluconeogenesis and lipogenesis may occur simultaneously in type 2

diabetes.

Although authors investigated using a mathematical model, causal relationship remains unclear. However, this study is of considerable interest.

Several points need to be clarified as follows:

1. The importance of mTORC1 and FoxO1 has been pointed out as a mechanism of selective insulin resistance (Proc. Natl. Acad. Sci. U. S. A. 107, 2010, 3441-3446; Cell Metab 23, 1154-1166, 2016). Are the authors able to investigate activation of mTORC1 or FoxO1 using the same mathematical model?

Answer: We had mTORC1, mTORC2, FoxO1, Srebp-1c and even the glucagon-PKA pathway in our original plan. However, we are limited by the availability of experimental data. The model, in its current form, is tailored to the two data sets used in the manuscript. In particular, we would need *in vivo* data as our main objective is the investigation of the impact of physiological insulin (glucagon) profiles, which are not available to our knowledge. We mentioned this additional layer of signaling as a possible extension of the model at the end of the discussion (line 548~554) and referred to the mentioned literature.

End of this answer.

2. Is expression levels of gluconeogenesis-associated genes such as PEPCCK and G6Pase regulated by Irs2-aPKC pathway?

Answer: Yes. There is evidence that aPKC exerts its role in limiting gluconeogenesis by blocking the activity of the CBP-CREB-CRTC2 complex, which is a co-transcription factor promoting the transcription of many gluconeogenesis enzymes, including PEPCK and G6Pase (He et al, Cell 136, 635-646, 2009). This is now mentioned in line 117 (citing He et al), line 402 and line 540.

It is widely believed that the Akt-FoxO axis is more potent than the aPKC-CREB axis in controlling gluconeogenesis. However, in the context of the insulin sensitive liver, Akt is not as responsive as aPKC to changes in the dynamic features of the postprandial insulin profile (Fig. 5). This allows us to propose a role of aPKC in separating the two time windows for glucose and lipid control instead of considering both Akt and aPKC here. We want to emphasize that this is different from insulin resistance in diabetes where Akt activation is impaired by different other mechanisms (line 534~537).

End of this answer.

3. Activation of IRS-1 and Akt seemed to be similar degree between pulsatile injection and T2D injection (Fig.2, Fig.4). However, in type 2 diabetes, lipogenesis was enhanced. Please explain this discrepancy in discussion.

Answer: Indeed, IRS1 and Akt activation level in pulsatile and T2D

injection shown in Fig. 2 and Fig. 4 are similar. The discrepancy regarding lipogenesis in T2D might be explained by considering our results shown in Fig. 5 and Fig. 6.

The “T2D” insulin infusion patterns (those in Fig. 2 and Fig. 4, now named T2D_a) do not mimic all the features of in vivo insulin. They mimic the 5-min pulse structure. However, the first-phase structure, namely a peak at about 30 min, is lacking. The red insulin curve in Fig. 5, also marked as “T2D” (we now renamed this pattern as T2D_b), has the impaired first-phase structure while lacking the 5-min pulse structure. The insulin profile in the right column of Fig. 6 (now named T2D_c) has an intact first-phase structure while the 5-min pulse structure is impaired. The signaling results in Fig. 6 and Fig. S4 support a role of aPKC hyperactivation in inducing hepatic lipogenesis.

We apologize for the confusion caused by labeling two different insulin profiles both as “T2D”. We now label the insulin profile in Fig. 2 and Fig. 4 as T2D_a, in Fig. 5 as T2D_b, and in Fig. 6 as T2D_c. We explain their difference in line 311~313 and 476~479.

End of this answer.

Reviewer #2 (Remarks to the Author):

The authors developed a mathematical model of the early cascade of hepatic insulin signalling. Two experimental data sets in rats allowed tuning the model, which is then used to simulate the time course of measured and unmeasured variables in response to different pattern of insulin stimulation. Based on model predictions, the most relevant feedback/crosstalk mechanisms among those reported in the literature are identified, and interesting hypothesis on the mechanisms underlying the more efficacious hepatic insulin response to pulsatile insulin delivery are formulated. The paper is well organized and clearly written, but some aspects of the modeling process appear to be weak, or at least are not clearly explained/motivated. This is particularly important in the present study given the limited amount of available data which strongly affect the reliability of quantitative model predictions.

Major

1. The model is not discussed in the context of previous literature, where several models of insulin signalling have been published. I invite the authors to compare their model with state-of-the-art-models, and to elaborate on the rationale underlying the choice of the state variables they included in their model.

I also point out to the authors' attention the recent paper by Satin et al, Mol Aspects Med 2016, reporting simulations of a key model of insulin receptor signaling during continuous and pulsatile insulin stimulations.

Answer: Thanks for pointing us to the new results and our drawbacks in the manuscript, which led us to re-evaluate our model. We previously compared our model with only one published hepatic insulin signaling model (line 419~426).

We now introduce the key insulin signaling model (the Sedaghat model, line 369~372) and explained the logic behind the way of our model design (line 373~386, 148~172 and 192~195). We didn't follow the Sedaghat model since it was largely based on *in vitro* experiments with adipocytes, which differ from hepatocytes in many aspect (discussed in line 373~382). An introduction of results from Satin et al and a comparison of the results are now provided in the discussion (line 485~497). Concerning the results in Satin's paper, it appears that their results and those from our model could be unified into a single mechanism, namely a non-linearly twisted dose-response curve, although qualitative differences exist. We propose that these two mechanisms might be complementary in different contexts (line 485~497).

End of this answer.

2. The proposed model has 39 parameters, to be quantified based on 15 samples, related to 4 different patterns of insulin stimulation. As the authors acknowledged, the model is obviously not uniquely identifiable, thus in general terms the reliability of parameter estimates is questionable, as well as the predictions derived from them. This raises doubts on the biological hypothesis based on model predictions, particularly of aPKC profiles, which are not supported by any data at all.

Answer: Thanks for pointing out this weakness in the manuscript. The lack of experimental data is an obvious limitation that can only be addressed by keeping the model complexity as low as possible and taking it into account for model selection. Hence, we 1) simplified the model to now 29 parameters in the minimal version; 2) introduced the Akaike information criterion with correction (AICc) in ranking models with different feedback structures; and 3) further added the additional constraint that the model results are robust against insulin variations. In other words, we computed AICc values based on model response to different insulin levels and used the mean AICc for model/parameter selection. These results are summarized in Table 1, line 236~242.

In addition, we confirm that that the two feedbacks in the minimal model are essential as removing any one of them from the full model led to unacceptable fitting results (Table 1, line 236~242). The main

statements derived from the model are robust across different model structures which, in our view, makes the predictions credible.

The main mechanism suggested by our modeling results was the aPKC switch, which lacks direct support by experimental data, as you emphasized. However, there exist some hints that might indirectly support fast switch-on and switch-off of aPKC after feeding, which is included in the discussion (line 440~454).

End of this answer.

3. By using a genetic algorithm, 50 points in the 39-dimension parameters space able to provide a reasonable fit were identified, but the information on how these points were selected is missing. No information is given on how the errors affecting the measurements are handled. I thus invite the authors to explain in the method section the identification procedure they adopted as well as the criteria they used to assess the validity of the estimates. A mandatory criterion is the biological plausibility of estimated values, in light of representative values which are available in the literature for most parameters.

Answer: Now we make each step clear (line 228~236 in the main text; page 7~8 in the supplement). The parameter sets are chosen based on mAICc, and 2 units difference in mAICc is used for model selection (Table 1 in the main text; page 7~8 in the supplement). The errors

affecting the measurements are included in the cost function for the numerical optimization (page 7~8 in the supplement). The requirement that the model response is robust to insulin variation, excludes many alternative solutions that are good in terms of AICc.

We found it difficult to directly compare our parameters with those reported by other models. The Sedaghat model, as well as the Koschorreck-Gilles model (Koschorreck, M., and Gilles, E.D. (2008) BMC Systems Biology 2, 43.), were largely based on *in vitro* studies of adipocytes, which differ in many aspects from hepatocytes (line 372~382). More importantly, parameters for receptor binding and unbinding in the Sedaghat model, as well as the Koschorreck-Gilles model, were validated for very low receptor concentration (0.1 nM, see Wanant S and Quon M 2000 J Theor Biol205(3): 355-364) while the receptor concentration for hepatocytes is much higher (40 nM, Rother, K.I. et al., (1998) J. Biol. Chem. 273, 17491–17497). Evidently, an exhaustive search of the whole admissible parameter space is impractical. However, the results presented have now been tested against different model structures (see previous point) and the reported values are able to generate robust response against variations in the level of insulin.

End of this answer.

4. In the present paper, only the 30 min AKT data from the study by Matveyenko et al. are used in model fitting, while the early data on IRS1 and IRS2 activation, available from the same study, are excluded. The authors justify their choice, writing that: “expected insulin profile might be established after some transient flow instability” which is rather cryptic to me. Given that the receptor activation is very fast, and that the model considers the early cascade, information on the dynamics of the involved substrates can be obtained only by early samples. Luckily, they are available in the Matveyenko et al. study, so why don't use them?

Answer: This is right, and we also thought about this. The study by Matveyenko et al. reported early time (<10 min) points of IRS1, IRS2, and Akt activation; however, the insulin concentration was not reported. Instead, the pumping rates of insulin delivery were schematically presented in Fig. 2 in their paper. The pre-hepatic insulin concentration in the canine studies in the same paper was reported in Fig. S1 in the supplement of Matveyenko et al. The portal vein insulin, measured at 1-minute intervals in the canine studies, showed that the short-term fluctuations in the portal vein insulin associated with different infusion patterns are large (Fig. S1 in Matveyenko et al). For example, in the case of constant insulin delivery, the insulin profile in the first 10 min decreased from 180 pM (measured at 1 min) to 60 pM

(measured at 10 min). For the T2DM pattern, insulin concentration started at the peak level while for the pulsatile pattern, insulin concentration started from the basal level. These observations prevented us from using the early samples (<10 min) of IRS1, IRS2 and Akt for model fitting. We now mention these early time point data in the study of Matveyenko et al in the manuscript (line 219~222) and explain, why we ignored them in the model analysis.

End of this answer.

5. Based on model predictions, the " feedback/crosstalk mechanisms that are indispensable for the model to fit the data" are selected. Can the authors please explain the criteria they implement to compare the alternative model structures, and to select the one they call the minimal one?

Answer: These steps are now clearly elaborated in Table 1, where we list all the models we have tested. The number of parameters, fitting performance as assessed by AICc and robustness to input variation as assessed by mAICc of each model are included in the table. Detailed explanation of those feedbacks is in line 174~190. Starting from the full model, we remove feedbacks one by one and finally reach the minimal model. We confirm that the two feedbacks in the minimal model are essential as removing any one of them from the full model

led to unacceptable fitting results (Table 1 and line 236~242).

End of this answer.

Minor

1. References in the Supplementary Information are labelled with numbers, and it is not clear to which item they refer to.

Answer: It's now corrected.

End of this answer.

2. Figure S2 shows distribution (ranges?) of fitted parameters, but labels do not always correspond to the symbols used in Model section of Supplementary Information. Please check.

Answer: It's now corrected. Please note that we need scaling factors (Sc1 for IRS1, Sc2 for IRS2 and Sc3 for Akt) as the measurements by western blot come without unit. Thus, the scaling factor for the refeeding experiment (with suffix e4) differs from that for the infusion experiments (with suffix e1). All other parameters have suffix e1.

End of this answer.

3. If I understand correctly, the boxes in figure S2 represent the mean +/- SD intervals. Thus I'd expect they are symmetrical with respect to

the mean, which is not the case for a few of them (e.g. e0p1e1). Please explain.

Answer: Sorry, this was unclear. Each parameter in this figure is normalized such that the median value is 1 (red). The boxes in this figure (marked by green) are upper and lower quartiles. Mean +/- SD intervals are presented along the labels of the vertical axis.

End of this answer.

4. I found some difficulties in reading the model equations in the Supplementary Information. Introducing some additional symbols in Figure 1, and the definition of symbols before the equations (and not at the end of the paragraph) could help the reader.

Answer: We now put definition of symbols before the equations. The model file (.txt file, provided as a single file) in the supplement has been optimized for reading.

End of this answer.

5. The pulsatile insulin input is modeled by a sinusoidal function, which is an approximation of the input used in the Matveyenko et al. study. The approximation is likely to introduce minor approximations on model simulations, but I don't see the rationale for not using the real profile.

Answer: We were forced to use such an approximation because the real insulin profile was not reported in this paper.

End of this answer.

Reviewers' comments:

Reviewer #1 (Remarks to the Author):

The authors have done a satisfactory job in addressing the previous concerns of this reviewer.

Reviewer #2 (Remarks to the Author):

The paper has been revised with a more careful discussion of the literature and of the rationale underlying the choice of model structure and state variables. Interestingly, the introduction of quantitative criteria to rank models with different feedback loops allowed the authors to limit model complexity. Some details on the identification procedure are now provided, but I still have some major concerns on some intrinsic limits of the model, related to the high number of model parameters to be estimated (29) compared to the small number of data (15).

Major

1. Back to my comment (point 2 of the previous review) on the reliability of parameter estimates, and thus of model predictions, the authors addressed this point by adopting, in the revised version of the ms, a simpler model, by using AIC to select acceptable solutions and then by filtering them based on the robustness to insulin input. This can help, but does not solve the problem. The model is obviously non identifiable, thus before performing parameter estimation by some algorithm, it should be important to understand which parameters or parameter aggregates are identifiable from the data and which are not. If I understand correctly, Figure S2 depicts some solutions (out of 50) that are able to provide similar fits to available data and are robust to insulin input data. For three parameters the range is wide and for most parameters the range is very narrow: this is surprising to me, since the limited number of data should not allow to obtain precise estimate for most parameters. In my opinion, Figure S2 represents some of the infinite solutions in the parameter space and some inconsistencies on RMS values in table 1 support this point (see point 3). A fitness landscape analysis could be of help for a better understanding of these solutions. Summing up, I invite the authors to properly address the identifiability issue: if Figure S2 summarizes all the possible solutions in the parameter space, it must be explicitly written and proved. If not, the limitations of the study should be clearly and explicitly discussed, namely since:

- the number of experimental data is very limited;
- the model is complex and clearly non identifiable from them;
- as far as I understand from the rebuttal no information is available to fix a priori some parameter values and/or to check the reliability of the estimated parameter values;

the proposed model represents a plausible, i.e. compatible with the data, description, but this does not mean that it is credible: if model parameters are not well (reliably) determined, the predicted dynamics are also not. Thus, the model can be used as a tool for generation and testing of hypothesis, but the limited credibility of the model obviously affects the credibility of the hypothesis.

2. Let's consider now the solutions identified by the genetic algorithm and shown in figure 2. Since they all compatible with the experimental data, the question is now: do these available solutions provide similar/consistent predictions of model variables for which data are not available? I presume that all model predictions examined in the ms and shown in figure 2,4,5,6 were calculated according to the mean parameter values. Since model predictions play a key role in hypothesis generation, it is mandatory to propagate the uncertainties affecting the parameter values identified by the algorithm, e.g. by showing in the figures the range of predictions resulting from the range of available solutions.

3. Finally, I have some concerns on some numbers shown in Table 1, namely: RMS decreases going from M1 to M2 even if M2 has a less rich structure than M1!! I expect that, moving from M1 (richer structure) to M2 (less rich structure) the model ability to fit the data worsens and thus RMS

increases. In my opinion, these findings some the are related to the existence of local solutions in the parameter space.

Minor

Please provide a reference for the adopted formulation of the Akaike criterion.

Reviewer #1 (Remarks to the Author):

The authors have done a satisfactory job in addressing the previous concerns of this reviewer.

Reviewer #2 (Remarks to the Author):

The paper has been revised with a more careful discussion of the literature and of the rationale underlying the choice of model structure and state variables. Interestingly, the introduction of quantitative criteria to rank models with different feedback loops allowed the authors to limit model complexity. Some details on the identification procedure are now provided, but I still have some major concerns on some intrinsic limits of the model, related to the high number of model parameters to be estimated (29) compared to the small number of data (15).

Major

1. Back to my comment (point 2 of the previous review) on the reliability of parameter estimates, and thus of model predictions, the authors addressed this point by adopting, in the revised version of the ms, a simpler model, by using AIC to select acceptable solutions and then by filtering them based on the robustness to insulin input. This can help, but does not solve the problem. The model is obviously non identifiable, thus before performing parameter estimation by some algorithm, it should be important to understand which parameters or parameter aggregates are identifiable from the data and which are not. If I understand correctly, Figure S2 depicts some solutions (out of 50) that are able to provide similar fits to available data and are robust to insulin input data. For three parameters the range is wide and for most parameters the range is very narrow: this is surprising to me, since the limited number of data should not allow to obtain precise estimate for most parameters. In my opinion, Figure S2 represents some of the infinite solutions in the parameter space and some inconsistencies on RMS values in table 1 support this point (see point 3). A fitness landscape analysis could be of help for a better understanding of these solutions. Summing up, I invite the authors to properly address the identifiability issue: if Figure S2 summarizes all the possible solutions in the parameter space, it must be explicitly written and proved. If not, the limitations of the study should be clearly and explicitly discussed, namely since:

- the number of experimental data is very limited;*
 - the model is complex and clearly non identifiable from them:*
 - as far as I understand from the rebuttal no information is available to fix a priori some parameter values and/or to check the reliability of the estimated parameter values;*
- the proposed model represents a plausible, i.e. compatible with the data, description, but this does not mean that it is credible: if model parameters are not well (reliably) determined, the predicted dynamics are also not. Thus, the model can be used as a tool for generation and testing*

of hypothesis, but the limited credibility of the model obviously affects the credibility of the hypothesis.

Answer: Thank you for raising this important point again, which forced us to look into more details of the fitting results and carried out a deeper identifiability analysis (line 596~625, supplement line 136~166). The new identifiability analysis, which has generated more diverse solutions, indicated that 11, out of 25 parameters, were not identifiable (line 247~251). This implies that these parameters cannot be interpreted quantitatively. Furthermore, your reasoning on the inconsistent RMS values in models of different complexity is correct (your point 3 below) and suggests that parameter fitting did not detect the global minimum (line 407~415). This reduces the reliability of the model predictions, as you state. In order to (partially) overcome this limitation we have used additional criteria in order to further classify the possible solutions (line 596~625, supplement line 136~166). We think, that despite the aforementioned limitations, the restriction of parameter solutions to a set that fulfills those additional criteria plus the side condition that all derived model predictions are consistent throughout all parameter solutions in this condensed set increases the credibility of the predictions. We clearly state the limitation in the Results (line 247~251) and the Discussion (line 407~411) and also explain our strategy in the Experimental Procedure (line 596~625) which increases the confidence that our predictions are worth being tested in further experiments.

Some more detailed remarks/explanations to this same point:

We would like to explain more about Fig. S2, which seemed to have caused some confusion. Fig. S2, in the 1st revision, depicts all the 8 solutions resulting from the mAICc criterion, which evaluates the robustness against input variation. In the original manuscript, we had provided 49 solutions, which had been only visually (not quantitatively) checked for robustness against input variation. This was not clearly described in the 1st revision.

Further, Figure S2, in the 1st revision, induced the impression that most of the parameters were identifiable, which is not the case, as confirmed by the extended analysis. In the 2nd revision (the current one) we updated Fig. S2 with the new results from the identifiability analysis (13 solutions, Fig. S1(a)). Moreover, we added two more figures (Fig. S1(b) and (c)), describing the solutions resulting from each step during the identifiability analysis, which facilitate understanding how the solutions were generated.

As mentioned above, we agree with the reviewer that the solutions we got are likely local minima in the parameter space, since the algorithm only converges to the global optimum when the run-time approaches infinity while in practice the algorithm is terminated by some empirical rules (supplement line 163~166). This is exactly the reason why we filtered out solutions by the relative distance criterion (line 615~617, supplement line 124), because we tried to avoid

analyzing solutions that reside around the same local minimum. In other words, the multiple solutions we got are different local minima that are not only acceptable in terms of fitting quality, but are also robust to input variations. Interestingly, the predictions are robust over all the solutions (Fig. S4, S5 and S6).

We would like to point out that our strategy of using the qualitative behavior of the model solutions in comparison to experimental constraints was not successful in determining parameter values but was successful in reducing the set of possible solutions to a degree that allowed to make consistent predictions within this set. This is not a guarantee but improves the credibility of the resulting hypothesis (line 596~625).

2. Let's consider now the solutions identified by the genetic algorithm and shown in figure 2. Since they all compatible with the experimental data, the question is now: do these available solutions provide similar/consistent predictions of model variables for which data are not available? I presume that all model predictions examined in the ms and shown in figure 2,4,5,6 were calculated according to the mean parameter values. Since model predictions play a key role in hypothesis generation, it is mandatory to propagate the uncertainties affecting the parameter values identified by the algorithm, e.g. by showing in the figures the range of predictions resulting from the range of available solutions.

Answer: We thank you for pointing out the importance of model variables for which data are not available. Following this comment, we looked into more details of the dynamics of insulin receptor and aPKC. Whereas aPKC always showed a bistable switch, insulin receptors showed two different patterns in the simulation of infusion experiments (Fig. S7): fast or slow internalization/recycling. As former experimental results support fast receptor internalization/recycling, we discarded solutions associated with slow receptor internalization/recycling (line 612~614, supplement line 151~157). This new finding, however, doesn't change model predictions (see below).

Fig. 4, 5 and 6 in the manuscript, which constitute the main predictions of the model, are computed with the **best** fitting solution, not with the mean solution. In this revision, the graphs have been updated by the results from the extended identifiability analysis. The updated Fig. S4, S5 and S6 show similar results that are computed with all the 13 solutions found by the identifiability analysis. Specifically, in Fig. S4(a), we show model response to three insulin-doses based on the 13 solutions, altogether 39 curves, which makes this figure rather complex. To facilitate understanding, we plotted in Fig. S4 (b) and (c) the critical information of the corresponding 39 curves in Fig. S4(a). We plotted in Fig. S5 (a) and (b) the critical information of model response to three 1st phase insulin profiles, for all the 13 solutions. In Fig. S6(b), we

showed all the 13 curves associated with the 13 solutions in response to pulsatile insulin. These figures support the robustness of model predictions and reveal the quantitative difference in the predictions associated with the 13 solutions.

3. Finally, I have some concerns on some numbers shown in Table 1, namely: RMS decreases going from M1 to M2 even if M2 has a less rich structure than M1!! I expect that, moving from M1 (richer structure) to M2 (less rich structure) the model ability to fit the data worsens and thus RMS increases. In my opinion, these findings some the are related to the existence of local solutions in the parameter space.

Answer: We agree that the inconsistency you pointed out is likely associated with local minima, due to the aforementioned reasons concerning the algorithm. We discussed how we dealt with the associated limitation in the response to point 1 above.

Minor

Please provide a reference for the adopted formulation of the Akaike criterion.

Answer: We have added a reference (line 230).

REVIEWERS' COMMENTS:

Reviewer #2 (Remarks to the Author):

Nothing to add: the authors have carefully responded to all my comments and satisfactorily revised the manuscript.